# Therapeutic Intervention with Anti-Complement Component 5 Antibody Does Not Reduce NASH but Does Attenuate Atherosclerosis and MIF Concentrations in Ldlr-/-.Leiden Mice

**DOI:** 10.3390/ijms231810736

**Published:** 2022-09-14

**Authors:** Florine Seidel, Robert Kleemann, Wim van Duyvenvoorde, Nikki van Trigt, Nanda Keijzer, Sandra van der Kooij, Cees van Kooten, Lars Verschuren, Aswin Menke, Amanda J. Kiliaan, Johnathan Winter, Timothy R. Hughes, B. Paul Morgan, Frank Baas, Kees Fluiter, Martine C. Morrison

**Affiliations:** 1Department of Metabolic Health Research, Netherlands Organisation for Applied Scientific Research (TNO), 2333 CK Leiden, The Netherlands; 2Department Medical Imaging, Anatomy, Donders Institute for Brain, Cognition, and Behavior, Radboud University Medical Center, 6525 EZ Nijmegen, The Netherlands; 3Department of Internal Medicine (Nephrology) and Transplant Center, Leiden University Medical Center, 2333 ZA Leiden, The Netherlands; 4Department of Microbiology and Systems Biology, Netherlands Organisation for Applied Scientific Research (TNO), 3704 HE Zeist, The Netherlands; 5Complement Biology Group, Systems Immunity Research Institute, School of Medicine, Cardiff University, Cardiff CF14 4XN, UK; 6UK Dementia Research Institute Cardiff, School of Medicine, Cardiff University, Cardiff CF24 4HQ, UK; 7Department of Clinical Genetics, Leiden University Medical Center, 2333 ZA Leiden, The Netherlands

**Keywords:** complement system, complement component 5, diet-induced, obesity, atherosclerosis, NASH, inflammation, therapeutic, macrophage migration inhibitory factor

## Abstract

Background: Chronic inflammation is an important driver in the progression of non-alcoholic steatohepatitis (NASH) and atherosclerosis. The complement system, one of the first lines of defense in innate immunity, has been implicated in both diseases. However, the potential therapeutic value of complement inhibition in the ongoing disease remains unclear. Methods: After 20 weeks of high-fat diet (HFD) feeding, obese Ldlr-/-.Leiden mice were treated twice a week with an established anti-C5 antibody (BB5.1) or vehicle control. A separate group of mice was kept on a chow diet as a healthy reference. After 12 weeks of treatment, NASH was analyzed histopathologically, and genome-wide hepatic gene expression was analyzed by next-generation sequencing and pathway analysis. Atherosclerotic lesion area and severity were quantified histopathologically in the aortic roots. Results: Anti-C5 treatment considerably reduced complement system activity in plasma and MAC deposition in the liver but did not affect NASH. Anti-C5 did, however, reduce the development of atherosclerosis, limiting the total lesion size and severity independently of an effect on plasma cholesterol but with reductions in oxidized LDL (oxLDL) and macrophage migration inhibitory factor (MIF). Conclusion: We show, for the first time, that treatment with an anti-C5 antibody in advanced stages of NASH is not sufficient to reduce the disease, while therapeutic intervention against established atherosclerosis is beneficial to limit further progression.

## 1. Introduction

Obesity is associated with a large range of cardiometabolic diseases, resulting in high morbidity and mortality worldwide [1]. Studies have shown that obesity is particularly associated with increased risk of developing non-alcoholic fatty liver disease (NAFLD) and atherosclerosis [2,3]. In both diseases, chronic inflammation, mostly triggered by an overload of dietary lipids, is known to play a critical role in driving disease progression [4,5]. Inflammation is crucial in the progression from simple steatosis to non-alcoholic steatohepatitis (NASH) in the liver, and in the vasculature, it promotes atherosclerotic plaque growth.

While the inflammatory processes at play in NAFLD and atherosclerosis remain partly unknown, several lines of evidence suggest the involvement of complement system activation. The complement system is one of the first lines of defense in innate immunity, activated in a sequential manner through one of three major pathways (alternative pathway, classical pathway or lectin pathway), which converge at complement component 3 (C3) activation. This subsequently leads to the activation of complement component 5 (C5), which initiates terminal pathway activation with the formation of the terminal complement complex (C5b-9 terminal complement complex, also known as membrane attack complex (MAC)) [6]. While the complement system is most well-known for its role in host defense, recently, attention has also been brought to its role in (sterile) metabolic inflammatory processes. Several studies have shown that circulating complement components (C3 and its activation product C3a) are increased in NAFLD patients relative to healthy controls [7,8,9]. In addition, the terminal complement system has been shown to be activated in NAFLD/NASH patients, leading to hepatic accumulation of MAC [10]. This complement activation has been proposed to be associated with hepatic immune cell infiltration as well as fibrogenesis [11,12], suggesting a role of complement activation in progressive NASH. A study that investigated the effect of knockout of C5 has provided further indication for a causal role of complement in NASH pathogenesis. This study demonstrated an attenuation of liver MAC deposition, steatosis and inflammation in the absence of C5 [13]. In a similar way, complement activation has also been linked to atherosclerosis progression. Circulating C3 has repeatedly been shown to be associated with cardiovascular disease risk (reviewed in [14]), and serum C5b-9 has been linked to atherosclerotic plaque stability in patients with acute ischemic stroke [15]. The involvement of complement activation in the atherosclerotic process has further been illustrated by the presence of C5 and MAC in human atherosclerotic plaques [16,17,18]. Complement system activation is thought to stimulate immune cell infiltration in atherosclerotic lesions and thereby contribute to disease progression [19,20]. Animal studies in models with knockout or overexpression of various complement components have provided further evidence for causal involvement of complement activation in atherosclerosis [21,22,23,24]. In addition, several investigators have shown that inhibitors of complement activation can reduce lesion growth in a preventive setting [25,26,27,28]. Wu et al., for instance, showed that preventive treatment with an anti-C5 antibody attenuated the development of atherosclerotic lesions in mCd59ab-/-/Apoe-/- mice [29]. However, it is still unclear whether the inhibition of terminal complement activation has therapeutic potential in established disease, with treatment starting when NASH or atherosclerosis is already present, as is inevitably the case in patients. In this study, we investigated, for the first time, the effect of limiting terminal complement activation on both NASH and atherosclerosis in one model simultaneously, by treatment with an established anti-C5 blocking monoclonal antibody (BB5.1) [30]. We studied its effect in Ldlr-/-.Leiden mice, a translational diet-induced mouse model for NASH and atherosclerosis. On a translational high-fat diet (HFD) with a macronutrient composition and cholesterol content comparable to human diets, Ldlr-/-.Leiden mice develop NASH and atherosclerosis in the context of an obese phenotype with insulin resistance, dyslipidemia and hypertriglyceridemia as typical for many patients [31,32,33]. This contrasts many other rodent models for NASH, which require harsh dietary conditions (e.g., methionine and choline deficiency or supraphysiological levels of cholesterol supplementation) and/or liver toxins (e.g., streptozotocin or carbon tetrachloride) to induce liver pathology and phenotypically do not reflect NASH patients well (e.g., lacking obesity, insulin resistance and/or dyslipidemia) [34]. Ldlr-/- mice also reflect underlying disease processes observed in NASH patients, as demonstrated by their concordance with various human transcriptomic [32,33,35,36] and plasma metabolomics disease profiles [33,37]. The dyslipidemic phenotype of Ldlr-/-.Leiden mice (with increased triglyceride and cholesterol levels, confined mainly to the atherogenic (V)LDL particles) results in the development of atherosclerosis in addition to NASH [32], thus, allowing the study of the effect of treatments on NASH and atherosclerosis simultaneously.

In the current study, we found that systemic inhibition of C5 activation did not affect the development of NASH but did limit further atherosclerotic lesion growth and severity independent of plasma cholesterol levels.

## 2. Results

To investigate the effect of an anti-C5 treatment (and thereby the inhibition of terminal complement pathway) in obesity, HFD-fed Ldlr-/-.Leiden mice were treated with an anti-C5 antibody (HFD+BB5.1) or vehicle control (HFD+vehicle) from week 20 to the end of the study (week 32, Figure 1). For reference, a separate group of mice was kept on a chow diet (Chow).

### 2.1. Anti-C5 Treatment Decreases Complement Activation

To confirm adequate inhibition of C5 by the employed dosage of the BB5.1 antibody in vivo, plasma samples were tested in a functional murine classical pathway ELISA, quantifying the amount of C9 deposition (Figure 2A). Before the start of treatment (week 20), complement activation in the HFD+vehicle and HFD+BB5.1 groups were comparable (HFD+vehicle 0.84 ± 0.05 arbitrary unit (AU) vs. HFD+BB5.1 0.75 ± 0.06 AU; not significantly different). After 2 weeks of treatment (week 22), the anti-C5 treatment strongly and significantly lowered terminal pathway activity (as shown by C9 deposition) in plasma relative to HFD+vehicle (HFD+vehicle 0.80 ± 0.06 AU vs. HFD+BB5.1 0.33 ± 0.02 AU, *p* ≤ 0.001), indicating that the antibody was present in plasma. This inhibitory effect persisted throughout the study and HFD+BB5.1 animals showed consistently lower levels of C9 deposition (all *p* ≤ 0.001) until the end of the study. Furthermore, in situ complement activation was assessed by determining the deposition of MAC, a C5-activation end product, in liver cross sections. This immunostaining revealed that at the end of the study, the anti-C5 treatment lowered the MAC-positive area (HFD+vehicle 25.7 ± 3.9 µm^2^/1000 µm^2^ vs. HFD+BB5.1 11.8 ± 2.0 µm^2^/1000 µm^2^, *p* ≤ 0.05, Figure 2B,C). Together, these results indicate an efficacious lowering of complement activation by the anti-C5 treatment.

### 2.2. Anti-C5 Treatment Does Not Affect HFD-Induced Obesity and Its Metabolic Risk Factors

From week 20 until the end of the study, HFD feeding was accompanied by a significantly higher fat intake in the HFD+vehicle group compared with Chow (Chow average 1.12 ± 0.02 kcal fat/day vs. HFD+vehicle group average 6.55 ± 0.09 kcal fat/day, *p* ≤ 0.001, Figure 3A), while HFD+vehicle and HFD+BB5.1 consumed the same amount of fat (average HFD+BB5.1 6.62 ± 0.12 kcal fat/day, not significantly different from HFD+vehicle). In HFD-vehicle animals, the average body weight was 54.9 ± 1.1 g at the end of the study, while the body weight of Chow animals was significantly lower and 35.4 ± 1.2 g in week 31 (*p* ≤ 0.001, Figure 3B). HFD+vehicle animals also displayed higher plasma total cholesterol and triglyceride concentrations compared to Chow (Chow average cholesterol 10.0 ± 0.4 mM, average triglycerides 1.7 ± 0.1 mM vs. HFD+vehicle average cholesterol 34.6 ± 2.1 mM, average triglycerides 5.7 ± 0.5 mM, *p* ≤ 0.001, Figure 3C,D). The anti-C5 treatment did not affect obesity development or the metabolic risk factors as the HFD+BB5.1 group displayed similar fat intake, body weight, plasma cholesterol and triglyceride concentrations compared to HFD+vehicle at all timepoints investigated (HFD+BB5.1 average cholesterol 33.5 ± 1.9 mM, average triglycerides 5.7 ± 0.7 mM, not significantly different from HFD+vehicle).

### 2.3. Anti-C5 Treatment Does Not Affect NASH Development

Histological examination of the livers revealed that HFD feeding induced the development of NASH as expected. HFD+vehicle animals developed marked steatosis (55 ± 3% of the cross-sectional area), including both pronounced macrovesicular (31 ± 2%) and microvesicular (25 ± 2%) steatosis, while Chow animals did not present any (0%) hepatic fat accumulation (*p* ≤ 0.001, Figure 4A–D). In HFD+vehicle animals, the steatosis was accompanied by an increased number of inflammatory aggregates, while inflammatory aggregates were practically absent in Chow (Chow 0.0 ± 0.0 inflammatory aggregates/mm^2^ vs. HFD+vehicle 2.9 ± 0.7 inflammatory aggregates/mm^2^, *p* ≤ 0.001, Figure 4E). Anti-C5 treatment did not affect liver histology as the levels of steatosis (52 ± 3%) (macrovesicular (27 ± 2%) and microvesicular (26 ± 3%)) and lobular inflammation 3.1 ± 1.3 inflammatory aggregates/mm^2^) were all comparable to HFD+vehicle (Figure 4A–E). HFD feeding also induced the development of fibrosis (Figure 4A). More specifically, the fibrosis area was increased in HFD+vehicle animals in comparison with Chow animals (0.1 ± 0.0% of cross-sectional area vs. HFD+vehicle 7.0 ± 2.9%, *p* ≤ 0.01, Figure 4F). The fibrosis stage was also more advanced for HFD+vehicle animals than Chow animals (Chow stage 1 ± 0. vs. HFD+vehicle stage 2 ± 0, *p* ≤ 0.05, Figure 4G). Anti-C5 treatment did not alter the fibrosis area (HFD+BB5.1 7.9 ± 4.1%, not significantly different from HFD+vehicle) or the fibrosis stage (HFD+BB5.1 2 ± 0, not significantly different). Furthermore, in line with the histology, anti-C5 treatment did not affect hepatic collagen content measured biochemically in tissue homogenates (HFD+vehicle 15.8 ± 2.5 µg/mg liver protein vs. HFD+BB5.1 16.6 ± 4.0, not significantly different, Figure 4H).

### 2.4. Anti-C5 Treatment Reverses HFD-Induced Effects on Oxidative Phosphorylation Genes

Next, we used Next Generation Sequencing (NGS), to assess whether anti-C5 treatment affected molecular processes associated with dysmetabolism and inflammation in NASH in the liver. Consistent with liver histology, NGS analyses revealed that HFD compared with Chow feeding significantly affected many (>300, all *p* ≤ 0.01) canonical pathways (full list shown in Appendix A), including many metabolic and inflammatory pathways related to NAFLD/NASH development. Among these significant pathways were ‘PPAR signaling’, ‘Mitochondrial Dysfunction’, ‘FCγ receptor-mediated Phagocytosis in Macrophages and Monocytes’, ‘NFκB Signaling ‘ and ‘Hepatic Fibrosis Signaling Pathway’ (all HFD+vehicle vs. Chow, *p* ≤ 0.01). In line with the observed effects on the histological level, anti-C5 treatment had very limited effects on hepatic gene expression, resulting in significant enrichment of only 3 pathways: ‘Oxidative Phosphorylation’, ‘Mitochondrial Dysfunction’ and ‘Sirtuin Signaling Pathway’. Importantly, the observed effect on oxidative phosphorylation was reversal of the HFD-induced effect. HFD feeding significantly impeded the oxidative phosphorylation pathway resulting in a significant predicted deactivation (HFD+vehicle vs. HFD+BB5.1 activation Z-score −6.4, *p* ≤ 0.001, Figure 5A), mainly due to pronounced effects on the mitochondrial complexes I and V. Notably, while many genes in the oxidative phosphorylation pathway were downregulated by HFD feeding, the anti-C5 treatment upregulated several genes and significantly activated this pathway (HFD+BB5.1 vs. HFD+vehicle activation Z-score 3.3, *p* ≤ 0.001, Figure 5B, full list of genes shown in Appendix A). Consistent with this, anti-C5 treatment also significantly affected the canonical pathway ‘mitochondrial dysfunction’ (HFD+BB5.1 vs. HFD+vehicle *p* ≤ 0.001). Altogether, these results indicate that anti-C5 treatment diminished the downregulation of genes triggered by HFD feeding and counteracted HFD-induced effects on the pathway ‘oxidative phosphorylation’.

### 2.5. Anti-C5 Treatment Reduces Atherosclerotic Plaque Growth and Severity Independently of Circulating Cholesterol

The effect of anti-C5 treatment on atherosclerosis development and lesion severity was assessed by histological analysis of cross-sections of the aortic root (valve area). The anti-C5 treatment significantly reduced the total lesion area (i.e., the atherosclerotic plaque load): HFD+vehicle animals had an average total lesion area of 693 ± 94 × 10^3^ µm^2^ per cross-section compared with 442 ± 50 × 10^3^ µm^2^ in HFD+BB5.1 animals (Figure 6A,B, *p* ≤ 0.05). More specifically, the anti-C5 treatment not only reduced plaque growth but also reduced the severity of the plaques. The HFD+BB5.1 group exhibited a larger area containing mild type I–III lesions compared with HFD+vehicle (HFD+vehicle 3 ± 1 × 10^3^ µm^2^ vs. HFD+BB5.1 10 ± 2 × 10^3^ µm^2^, *p* ≤ 0.05, Figure 6C). Conversely, the total area attributable to severe IV–V lesions was significantly smaller in the HFD+BB5.1 group than in the HFD+vehicle group (HFD+vehicle 684 ± 98 × 10^3^ µm^2^ vs. HFD+BB5.1 421 ± 52 × 10^3^ µm^2^, *p* ≤ 0.05, Figure 6D). This anti-atherogenic effect of the anti-C5 treatment was not associated with a reduction in MAC deposition within atherosclerotic plaques (Appendix A). These data indicate a beneficial effect of the anti-C5 treatment against atherosclerosis development that is not the result of terminal pathway inhibition within the plaque, but rather results from an indirect effect on a systemic pro-atherogenic factor, such as cholesterol, oxidized LDL (oxLDL) or inflammation. As plasma cholesterol levels were not different between HFD+vehicle and HFD+BB5.1 animals (Figure 3E), the effect was independent of cholesterol exposure. More specifically, atherosclerosis development in Ldlr-/-.Leiden mice is strongly driven by plasma cholesterol (i.e., cholesterol exposure, Figure 6E). Based on historical data on the close relationship between plasma cholesterol and total lesion area in Ldlr-/-.Leiden mice, the expected lesion area at the start of treatment was 446 × 10^3^ µm^2^ per cross-section (i.e., expected lesion area t = 20 weeks). This graph also shows that the observed lesion development in the HFD+vehicle group lies within the 95% prediction band, while the total lesion area in HFD+BB5.1 falls outside the 95% prediction band, further supporting the notion that the effect of the anti-C5 treatment is independent of an effect on plasma total cholesterol. Next, we analyzed plasma levels of oxLDL as a potential pro-atherogenic mediator. HFD+vehicle mice had significantly higher plasma oxLDL levels than Chow mice (Chow 464.6 ± 12.2 nmol/mg vs. HFD+vehicle 542.2 ± 18.9 nmol/mL, *p* ≤ 0.05, Figure 6F). The anti-C5 treatment reversed this HFD-induced increase (HFD+BB5.1 average 463.3 ± 17.8 nmol/mL, *p* ≤ 0.01 vs. HFD+vehicle), providing a potential rationale for the observed anti-atherogenic effect of BB5.1 treatment. Finally, we investigated the effect of the anti-C5 treatment on (vascular) inflammation as a potential additional mechanism. We first analyzed vascular adhesion molecules, which mediate immune cell infiltration in atherogenesis, and found that HFD increased the plasma levels of vascular cell adhesion molecule-1 (VCAM-1) (Chow 2.08 ± 0.09 µg/mL vs. HFD+vehicle 2.85 ± 0.08 µg/mL; *p* ≤ 0.001), intracellular adhesion mocelcule-1 (ICAM-1) (Chow 145.09 ± 6.84 ng/mL vs. HFD+vehicle 234.10 ± 10.32 ng/mL; *p* ≤0.001), E-selectin (Chow 56.86 ± 3.54 ng/mL vs. HFD+vehicle 70.15 ± 1.93 ng/mL; *p* ≤ 0.05) and P-selectin (Chow 223.12 ± 7.65 ng/mL vs. HFD+vehicle 350.84 ± 16.43 ng/mL; *p* ≤ 0.01). However, the anti-C5 treatment did not attenuate these inductions. Then, we analyzed chemotactic cytokines involved in immune cell recruitment during atherogenesis, monocyte chemoattractant protein-1 (MCP-1) and macrophage migration inhibitory factor (MIF). MCP-1 was induced by HFD feeding relative to Chow (Chow 26.34 ± 11.25 ng/mL vs. HFD+vehicle 44.56 ± 2.69 ng/mL; *p* ≤ 0.01) but was not lowered by the anti-C5 treatment (46.31 ± 5.14 ng/mL; not significant). In contrast, MIF—which correlated significantly with total atherosclerotic lesion area (*p* = 0.033, ρ = 0.47, Figure 7A)—was significantly reduced by the anti-C5 treatment (HFD+vehicle 49.1 ng/mL; HFD+BB5.1 34.6 ng/mL; *p* ≤ 0.05, Figure 7B).

## 3. Discussion

A growing body of evidence indicates that the terminal complement system plays a role in NASH and atherosclerosis progression. However, the therapeutic value of terminal complement system inhibition in already established disease has not been studied to date. In this study, we investigated the potential beneficial effect of anti-C5 treatment against obesity-associated NASH and atherosclerosis in a therapeutic study protocol (starting the treatment in already established disease). For this we used HFD-fed Ldlr-/-.Leiden mice, a translational model for obesity-associated NASH and atherosclerosis development [32,33]. Anti-C5 treatment in these mice did not have any effect on the development of obesity or associated metabolic risk factors and did not affect the development of steatosis, inflammation or fibrosis in the liver. It did, however, reduce the development of atherosclerosis in the aortic root, as shown by a reduced progression of total lesion area and a shift towards milder lesion types. While it has been hypothesized that complement inhibition may form a viable target for the treatment of NASH, we did not find any effect of anti-C5 treatment on liver steatosis and inflammation despite a significant inhibition of MAC deposition in the liver. This observation is not in line with a previous study showing that knockout of complement C5 in C57BL/6 mice attenuated HFD-induced development of hepatic steatosis and had some inflammation-reducing effects in the liver [13]. This discrepancy might be partly explained by differences in the experimental model used. NAFLD induction by short-term (10-week) HFD feeding in C57BL/6 mice is much milder than in the current study (e.g., mild steatosis; very early—not histologically quantifiable—inflammation; no fibrosis; ALT levels well within the normal range) and in absence of the metabolic dysfunction phenotype (obesity, insulin resistance, dyslipidemia) that characterizes most NASH patients. Furthermore, the anti-C5 treatment in our study was started once NASH was already established (i.e., after 20 weeks of HFD feeding), while C5 knockout is by definition preventive. The transition from benign steatosis to NASH is considered to involve neutrophil accumulation as a key mechanism [38] and studies have shown that upon activation of the terminal complement cascade, the complement fragment C5a (obtained from the cleavage of C5) enhances neutrophil survival in inflammatory states by inhibiting neutrophil apoptosis [39]. In Ldlr-/-.Leiden mice, inflammation is already present at 6 weeks of HFD feeding and steatosis, both macrovesicular and microvesicular, are shown to be maximal at approximately 20 weeks of HFD [36]. This suggests that, if the complement system is indeed involved in the transition from non-alcoholic fatty liver (NAFL) to NASH, the stage of the disease at the start of the treatment in the current study might have been too advanced to allow successful intervention. In addition, in the current study the degree of terminal complement system inhibition in the liver as assessed by inhibition of MAC deposition (which, although substantial, was not complete), may not have been large enough to lead to an overall attenuating effect on hepatic inflammation. All the more so, since inflammation in the context of NASH is known to be characterized by a broad activation of inflammatory pathways driven by multiple pro-inflammatory triggers [40,41]. Furthermore, it is possible that the antibody treatment in our study especially targeted systemic (circulating) complement and was less efficient in inhibiting local complement.

The absence of effect of the anti-C5 treatment on NASH was also reflected in the transcriptomics analysis, which did not show any effect of the treatment on NASH-related pathways for steatosis, inflammation and fibrosis. We did, however, observe an effect of the anti-C5 treatment on the canonical pathways ‘oxidative phosphorylation’ and ‘mitochondrial dysfunction’. Several findings suggest an interaction between complement system (including terminal pathway) and mitochondrial function. Martinus and Cook [42] demonstrated that in vitro exposure to C5a inhibits dehydrogenase and cytochrome c activities. Furthermore, the end product of terminal pathway activation MAC, which is mainly known for forming cytotoxic pores on target cells, has more recently been shown to have a range of other effects at sublytic levels. Sublytic MAC has been described to result in mitochondrial damage and release of cytochrome c in the cytosol [43]. Although mitochondrial dysfunction and related oxidative stress are known to play a crucial role in the development of NASH [44], the predicted beneficial effect on mitochondria and oxidative phosphorylation-related gene expression observed in our study seems not to be sufficient to improve liver histology. NASH is a complex multifactorial disease in which many different inflammatory triggers and pathways drive its progression, and the enhancement of mitochondria function solely was probably not sufficient to limit or reverse the progression of the disease.

In our study, we further investigated the effect of anti-C5 treatment on the development of atherosclerosis. We showed that anti-C5 treatment was beneficial against atherosclerosis by limiting the total lesion size and severity. These results are in line with previous knockout studies for complement system factors that also revealed that deficiency in C6, which is necessary for MAC formation by binding to C5b, protects against atherosclerosis in rodents [21]. Consistently, other studies have demonstrated that a knockout of the main negative regulator of MAC formation, CD59, accelerates the development of atherosclerotic lesions [22,45], while anti-C5 treatment in a preventive setting attenuates this development [29]. In our study, we specifically demonstrated that the use of an anti-C5 treatment in a therapeutic intervention (starting at a time point at which pronounced disease is already present) is also efficacious in slowing down further atherosclerosis progression. However, other studies showed that C5 knockout in ApoE-/- mice did not affect atherosclerotic lesion area [46]. This suggests that the effect of C5 inhibition on atherosclerosis progression may depend on the mouse model. ApoE has recently been revealed to bind complement factor C1q of the classical complement pathway resulting in the inhibition of this pathway [47]. Consequently, in ApoE-/- mice, increased activation of the classical pathway was shown to occur due to the lack of the ApoE–C1q complex. The implication of the upstream classical pathway independent of terminal pathway activation is unclear, and over activation of the classical pathway in absence of ApoE might counterbalance the beneficial effect of a C5 knockout on atherosclerosis.

In the Ldlr-/-.Leiden mouse model, we more specifically showed that atherosclerosis progression is driven by cholesterol exposure in such a way that a linear relationship can be made. While the HFD-control and the anti-C5 treated groups displayed comparable concentrations of plasma cholesterol, the atherosclerotic lesions in HFD-control animals seemed to have grown further from the start of the treatment and worsened towards stage IV and V. In comparison, the anti-C5 treated animals displayed a smaller lesion area overall, with a higher proportion of milder type-III lesions remaining. Thus, it seems anti-C5 treatment did not reduce the size of the already present atherosclerotic plaques but limited further growth and worsening of the plaques independent of cholesterol exposure. This anti-atherosclerotic effect of the anti-C5 treatment was not associated with a reduction in MAC deposition within the atherosclerotic plaques. This could be related to poor penetration of the vascular tissue by the antibody used [48] or related to the timing of the intervention, and indicates that the observed effects must be the result of an indirect effect of the treatment on systemic pro-atherogenic factors, such as cholesterol, oxLDL or inflammation.

While circulating cholesterol itself does not seem to explain the observed effect on atherosclerosis, oxLDL is also known to drive the progression of atherosclerotic plaques [49]. In HFD feeding, high circulating oxLDL resulting from increased oxidative stress (in the liver or in circulation [50]) can be absorbed by macrophages in the plaque resulting in their foamy appearance and crystal formation. Consistent with the findings by Dai et al., who showed that the inhibition of C3 activation reduced serum oxLDL [26], the anti-C5 treatment in the current study lowered circulating oxLDL levels, which may have contributed to the observed reduction in atherosclerosis progression. Furthermore, although the full implication of complement system activation in atherosclerosis is not understood, C5a has been shown in in vitro studies to induce adhesion molecule expression by endothelial cells [51] and to be a potent chemotactic for several inflammatory cells including monocytes [52], and could, therefore, play an important role in immune cell accumulation in the arterial wall [53]. In vivo, adhesion molecule expression is regulated by a broad range of pro-inflammatory mediators [54], most likely with a large degree of redundancy [55]. Our findings are in line with this notion, showing that anti-C5 treatment alone is not sufficient to lower adhesion molecule levels. We did observe a reduction in the levels of the pro-atherogenic chemotactic cytokine MIF. MIF has been shown to be expressed in mouse and human atherosclerotic lesions and is known to play a crucial causal role in atherogenesis, as demonstrated by genetic knockouts and immunoneutralization studies [56,57,58,59]. Consistent with this, we found a positive correlation between plasma MIF levels and total lesion load. The observed reduction in MIF but not MCP-1 is in line with studies reporting that C5a enhances the release of MIF in vitro and in vivo [59], while genetic ablation of C5 was shown not to affect circulating MCP-1 levels in vivo [60]. A mechanistic rationale involving MIF is further supported by the observation that oxLDL induces MIF in endothelial cells [57] and vascular smooth muscle cells [61]. Here, inhibition of C5, and subsequent C5a formation, may have reduced monocyte accumulation through reduction in oxLDL and MIF, preventing further inflammation and progression of the atherosclerotic plaque to more severe stages.

Altogether, we show for the first time that anti-C5 treatment in the ongoing disease process, when disease is already established, does not reduce the further development of NASH in an established translational preclinical disease model. In the vasculature, however, anti-C5 treatment does prevent further lesion progression, reducing total atherosclerotic lesion load and shifting lesion severity towards milder lesion types, in absence of an effect on plasma cholesterol.

## 4. Materials and Methods

### 4.1. Animal Studies

All animal experiments were conducted in accordance with the European Union regulations on animal research and approved by an independent Animal Welfare Body (IVD TNO; approval number TNO-451). Male Ldlr-/-.Leiden mice were bred at TNO Metabolic Health Research, Leiden, The Netherlands, and group-housed in a specific-pathogen-free animal facility (temperature ~21 °C, relative humidity 50–60%) on a 12-h light/dark cycle. The animals received food and water ad libitum. All mice were fed a standard chow diet (Sniff R/M V1530, Uden, The Netherlands) until the start of the study. At the start of the study (t = 0 weeks), 17–18 week-old mice were matched into three experimental groups based on body weight, blood glucose, plasma cholesterol and plasma triglyceride levels. One group (Chow, n = 8) remained on the standardized chow diet as a healthy reference. The other two groups (HFD+vehicle, n = 17 and HFD+BB5.1, n = 17) were fed a well-established obesity-inducing [62] energy-dense HFD (45 kcal% fat with 39 kcal% fat from lard and 6 kcal% fat from soybean oil, 20 kcal% protein and 35 kcal% carbohydrates, D12451, Research Diets, New Brunswick, NJ, USA). Mice were treated with a vehicle (sterile phosphate-buffered saline (PBS)) or the established anti-C5 antibody (BB5.1 [30]) twice a week from t = 20 weeks until the end of the study (t = 32 weeks). Based on previous in-house data the development of NASH and atherosclerosis histopathology was comparable between HFD controls treated with an IgG isotype and untreated HFD controls (Appendix A). Therefore, an IgG isotype control was not included in the study, but vehicle injections with PBS were included to control for the stress of handling and injecting the mice. Chow and HFD+vehicle animals received 200-μL intraperitoneal injections of PBS while HFD+BB5.1 received 200-μL intraperitoneal injections of BB5.1 anti-C5 antibody (5 mg/mL in PBS, 1 mg/mouse). The experimental scheme for the study is shown in Figure 1.

Group sizes were based on power calculations for detection of a difference in the main end-point hepatic inflammation (at α = 0.05 and power = 0.8), on top of this two animals per group were added to compensate for potential drop-outs during the study. Each animal cage contained two to six animals from the same experimental group. To provide equal housing conditions for each group and avoid bias, cages from different groups were randomized (using a Random Integer generator) over the shelves in the animal room. For administrating the treatments, the animals were injected per group to minimize the risk of a dosing error, but the order of the cages was rotated equally between all time points. Every week, individual body weight was measured, and food intake was monitored per cage to calculate the average food intake per mouse per day. Blood (200 μL) was collected via the tail vein at weeks 20, 22, 26 and 31 weeks after a 5-h fast. At t = 32 weeks, mice were sacrificed by isoflurane inhalation (4%) and heart puncture, and non-fasted terminal blood was collected. After that, mice were perfused with PBS for 10 min (1 mL/min) and the livers and hearts (including aortic root) were isolated. For the blood samplings and sacrifice, the order of the cages was randomized. In addition, all mice were individually numbered in a way that people conducting the experiment, as well as the analyses, were only aware of the mouse number and blinded to the corresponding experimental group. For fat intake analyses, each cage was considered an experimental unit, for all other experiments and analyses, single animals were considered to be an experimental unit. There were no drop outs during the study and all animals were used with no exclusions except where explicitly mentioned.

### 4.2. BB5.1 Antibody Production

The BB5.1-producing cell line was grown in CELLine CL1000 bioreactor flasks (Sigma-Aldrich, Zwijndrecht, The Netherlands) in RPMI medium containing 10% Ultra Low IgG Fetal Bovine Serum (Thermo Fisher Scientific, Waltham, MA, USA). The cell compartment was harvested every two weeks. Harvests were pooled, 0.2 μm filtered and applied to a Protein G FPLC column (Generon Ltd., Slough, UK) on an AKTA FPLC freshly washed to ensure sterility. After washing, bound IgG was eluted from the column as recommended and collected aseptically. Eluted IgG was dialyzed into PBS and stored in aliquots at −20 °C. All batches were tested and shown to be endotoxin-free before freezing. In addition, the batches were negative for mouse pathogens (based on MAP test performed by QM Diagnostics, Nijmegen, The Netherlands).

### 4.3. Plasma and Liver Biochemical Analyses

#### 4.3.1. Blood Glucose, Plasma Triglycerides and Cholesterol

Blood glucose was measured at the time of the blood collection using a glucometer (Freestyle Disectronic, Vianen, The Netherlands). Blood samples were used to prepare EDTA plasma by centrifugation (10 min, 6000 rpm). In freshly prepared plasma, total triglyceride and cholesterol concentrations were measured using enzymatic assays (triglyceride GPO-PAP and cholesterol CHOD-PAP, Roche Diagnostics, Almere, The Netherlands).

#### 4.3.2. Complement Component 9 (C9) Functional Complement Enzyme-Linked Immunoassay (ELISA)

A functional complement pathway ELISA that allows analysis of complement activation and subsequent C9 deposition was performed using plasma samples as described previously [63]. In short, to maximally trigger the activation of the classical complement pathway, NuncMaxisorp plates (Thermo Fisher Scientific) were coated with human IgM (in-house, LUMC, Leiden, The Netherlands) at 2 μg/mL in 0.1 M carbonate buffer overnight at room temperature. The plates were then blocked with PBS 1% bovine serum albumin (BSA) (1 h, 37 °C) and washed three times with PBS-Tween. The plasma samples were diluted with BVB++ buffer (Veronal buffered Saline/0.5 mM MgCl2/2 mM CaCl2/0.05% Tween 20/1% BSA, pH 7.5, all from Sigma-Aldrich) and incubated in the plates (1 h, 37 °C). CD1 NMS (Innovative Research, Novi, MI, USA) corresponding to 100 arbitrary units (AU) were used as standards for the assay. Plates were washed three times with PBS-Tween and incubated with digoxygenin-conjugated rabbit IgG anti-recombinant mouse C9 (1/200 in PBS-Tween 1% BSA, in-house, LUMC, Leiden, The Netherlands). The plates were washed three times again and incubated with Fab anti-digoxygenin-conjugated-POD (1/2500 in PBS-Tween 1% BSA, Roche Diagnostics) (1 h, 37 °C) and washed again three times. The plates were finally incubated with tetramethylbenzidine for between 10 and 30 min and the reaction was stopped with sulfuric acid. The optical density was measured at 450 nm with a spectrophotometer (iMark Microplate Reader, Bio-Rad, Hercules, CA, USA).

#### 4.3.3. Hepatic Collagen Content

Liver homogenates were prepared from sinister lobe biopsies using glass beads. Total collagen contents were measured in the liver homogenates after acid hydrolysis using the Sensitive Tissue Collagen Assay (Quickzyme, Leiden, The Netherlands). The measured collagen was normalized by the corresponding total protein content, which was measured in the same hydrolysates using the Total Protein Assay (Quickzyme).

#### 4.3.4. Plasma oxLDL, Chemokines and Adhesion Molecules

OxLDL was measured in plasma one week before sacrifice using an oxLDL ELISA kit according to manufacturer’s instructions (CSB-E07933m, CUSABIO, Houston, TX, USA). In the same plasma samples, MIF was quantified with the MIF DuoSet ELISA kit (DY1978, R&D Systems, Minneapolis, MN, USA). The concentrations of MCP-1 and adhesion molecules were analyzed in plasma as described previously [64]. Briefly, ELISAs were performed to measure MCP-1, E-selectin, P-selectin, ICAM-1 and VCAM-1 (with MCP-1 DuoSet (DY479-05), E-selectin/CD26E DuoSet (DY575), P-selectin/CD62P DuoSet (DY737), ICAM-1/CD54 DuoSet (DY796), VCAM-1/CD106 DuoSet, (DY643), respectively, all from R&D Systems).

### 4.4. Histological Analyses of the Livers and Aortic Roots

#### 4.4.1. NAFLD/NASH and Atherosclerosis

The livers (left lobe) and the aortic roots of the hearts were collected at sacrifice and fixed in buffered formalin (3.7%) for between 24 and 48 h. Samples were dehydrated overnight (Automatic Tissue Processor ASP300S, Leica Biosystems, Amsterdam, The Netherlands) and embedded in paraffin. Liver cross-sections (3 µm) were stained with hematoxylin-eosin or Sirius Red for NASH or fibrosis analyses, respectively. NASH and fibrosis were scored by a board-certified pathologist using an adaptation of the grading system for human NASH as described previously [33,65]. Briefly, macrovesicular, microvesicular and total steatosis were microscopically evaluated at 40× to 100× magnification and expressed as percentage of the liver cross-sectional area. Inflammation was assessed by the number of inflammatory aggregates per microscope field (4.4 mm^2^) using a 100× magnification. Fibrosis was assessed by the fibrosis area (expressed as the percentage of the liver cross-sectional area) and the fibrosis stage as reported previously [32]. For the NASH and fibrosis area analysis, 5 non-overlapping microscope fields were evaluated.

Cross-sections of the aortic roots (5 µm) were stained with hematoxylin-phloxin-saphran and scanned (Aperio Digital Pathology Slide Scanner AT2, Leica, Amsterdam, The Netherlands). Specimens in which the aorta was not perpendicular to the cross-sectional plane were excluded from analysis, resulting in n = 7 samples in the HFD+vehicle group and n = 12 samples in the HFD+BB5.1 group. Atherosclerotic lesions were analyzed, as described in previous studies [66,67], with a pathology slide viewer (Aperio ImageScope, Leica, Amsterdam, The Netherlands). Briefly, a total of four cross-sections (at 50 µm intervals) was analyzed for each animal, determining total lesion area per cross-section as well as lesion severity (according to an adapted scoring system of the American Heart Association [68,69]) Using this system, atherosclerotic lesions were classified into five types: 1: early fatty streak (lesion with up to 10 foam cells in the intima without any other change); 2: regular fatty streak (lesion with 10 or more foam cells in the intima without any other change); 3: mild plaque (foams cells in the intima and fibrotic cap present); 4: moderate plaque (infiltration of the lesion in the media, elastic fibers still intact); 5: severe plaque (severe disruption of the media with disrupted elastic fibers, presence of cholesterol crystals, calcium deposits and probable presence of necrotic core).

#### 4.4.2. MAC Immunostaining

Cross-sections of the liver (sinister lobe) and aorta roots were deparaffinized in xylene and rehydrated with alcohol gradients and demineralized water. Endogenous peroxidase was blocked with 30% hydrogen peroxide in methanol during 15 min and slides were subsequently washed in ethanol and demineralized water. Heat-induced epitope retrieval was performed with citrate buffer (pH 6) using a Dako PT-link device (65 °C, 97 °C, 65 °C, Dako, Glostrup, Denmark) over 10 min. After washing with 0.05% Tween 20 (Sigma-Aldrich) in PBS, the sections were blocked with 5% normal goat serum over 45 min at room temperature. The sections were incubated with rabbit anti-mouse C5b-9 antibody (ab55811, 1:1000 in PBS with 1% normal goat serum, Abcam, Cambridge, UK) overnight at 4 °C and washed three times with 0.05% Tween 20 in PBS. A secondary horseradish peroxidase-preabsorbed goat anti-rabbit antibody (ab97080, 1:800 in PBS with 1% normal goat serum and 0.05% Tween 20, Cambridge, UK) was incubated for 1 h at room temperature. The sections were washed again three times with 0.05% Tween 20 in PBS and incubated with diaminobenzidine (DAB substrate kit SK-4100 Vector Laboratories Inc., Burlingame, CA, USA) for 3 min. The sections were washed in tap water and demineralized water before being counterstained with hematoxylin. The sections were then rinsed for 10 min in tap water and dehydrated with successively demineralized water, alcohol gradients and xylene. Finally, the slides were digitalized using the Aperio digital slide scanner (Aperio AT2, Leica). An automated quantification of the immuno-positive area was performed in five non-overlapping fields for the liver cross-sections and in the atherosclerotic lesions for the cross-sections of the aortic roots using ImageJ (v2.1.0/1.53c; Java 1.8.0_172 [64-bit]).

### 4.5. Liver Gene Expression and Pathway Analysis

Ribonucleic acid (RNA) was extracted from 25–40 mg snap-frozen liver (sinister lobe, Chow n = 6, HFD+vehicle n = 12, HFD+BB5.1 n = 12) as previously described [70]. RNA integrity and concentration were examined for each sample using the RNA 6000 Nano LabChip kit and a bioanalyzer 2100 (both Agilent Technologies, Amstelveen, The Netherlands). Next Generation Sequencing was performed at Genome Scan (Leiden, The Netherlands). For this, strand-specific cDNA libraries were generated from RNA using oligo-dT magnetic beads, mRNA fragmentation, NEBNext Ultra Directional RNA Library Prep Kit for Illumina (New England BioLabs, Ipswich, MA, USA), NEB #E7420S/L resulting in 300–500 bp amplified libraries/sample. Libraries were multiplexed, clustered and sequenced on a NovaSeq6000 system for Illumina, (New England Biolabs, Inc., Ipswich, MA, USA), with a paired-end, 150 bp sequencing protocol, 20 million reads per sample and indexing. RNA counts were preprocessed and checked for integrity and quality as described [62]. Based on this quality control, one sample from the HFD+vehicle group was excluded from further analysis as a technical outlier. Differentially expressed genes were determined using the Deseq2-pipeline [71] using a statistical cut-off of *p* < 0.01 for the comparisons HFD+vehicle vs. Chow and HFD+BB5.1 vs. HFD+vehicle. These DEG were used as input for canonical pathway analysis using the Ingenuity Pathway Analysis suite (IPA; www.ingenuity.com, accessed on 25 August 2021) as detailed elsewhere [62,72,73].

### 4.6. Relationship between Cholesterol Exposure and Atherosclerotic Lesion Size

The relationship between cholesterol exposure and atherosclerotic lesion size in Ldlr-/-.Leiden mice was investigated using control groups from previously published data [32,67,74] and historical in-house data (unpublished). Cholesterol exposure (mM/week) was calculated as the plasma cholesterol concentration (mM) multiplied by the time during which the animals were exposed to this cholesterol concentration (weeks). The total atherosclerotic lesion area was determined as described above.

### 4.7. Statistical Analysis

All data are shown as means ± standard error of the mean (SEM). Statistical differences were determined between Chow and HFD+vehicle and between HFD+vehicle vs. HFD+BB5.1 groups using SPSS software (version 25, IBM, Armonk, NY, USA). For normally-distributed and homoscedastic or non-homoscedastic variables an analysis of variances (ANOVA) followed by a least significant difference (LSD) or a Dunnett T3 post-hoc test were performed, respectively, while for data that was not normally distributed a non-parametric Kruskal–Wallis test in addition to Dunn–Bonferroni multiple comparisons were chosen. Finally, for the atherosclerosis analyses in which only HFD+vehicle and HFD+BB5.1 were compared, a t-test for normally-distributed variables and a Mann–Whitney U test for not normally-distributed data were executed. Two-tailed *p*-value (P) were used and *p* ≤ 0.05 was considered significant.

## Figures and Tables

**Figure 1 ijms-23-10736-f001:**
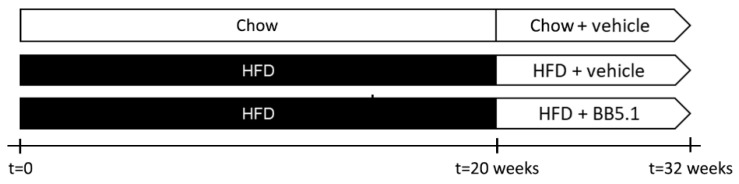
Experimental design. All mice were fed a standard chow diet until the start of the study. At the start of the study (t = 0 weeks), one group (Chow, n = 8) remained on the standardized chow diet as a healthy reference and the other two groups (HFD+vehicle, n = 17 and HFD+BB5.1, n = 17) were fed a well-established obesity-inducing energy-dense HFD. Twice a week, from t = 20 weeks until the end of the study (t = 32 weeks), Chow and HFD+vehicle animals received 200-μL intraperitoneal injections of PBS, while HFD+BB5.1 received 200-μL intraperitoneal injections of BB5.1 anti-C5 antibody (1 mg/mouse in PBS).

**Figure 2 ijms-23-10736-f002:**
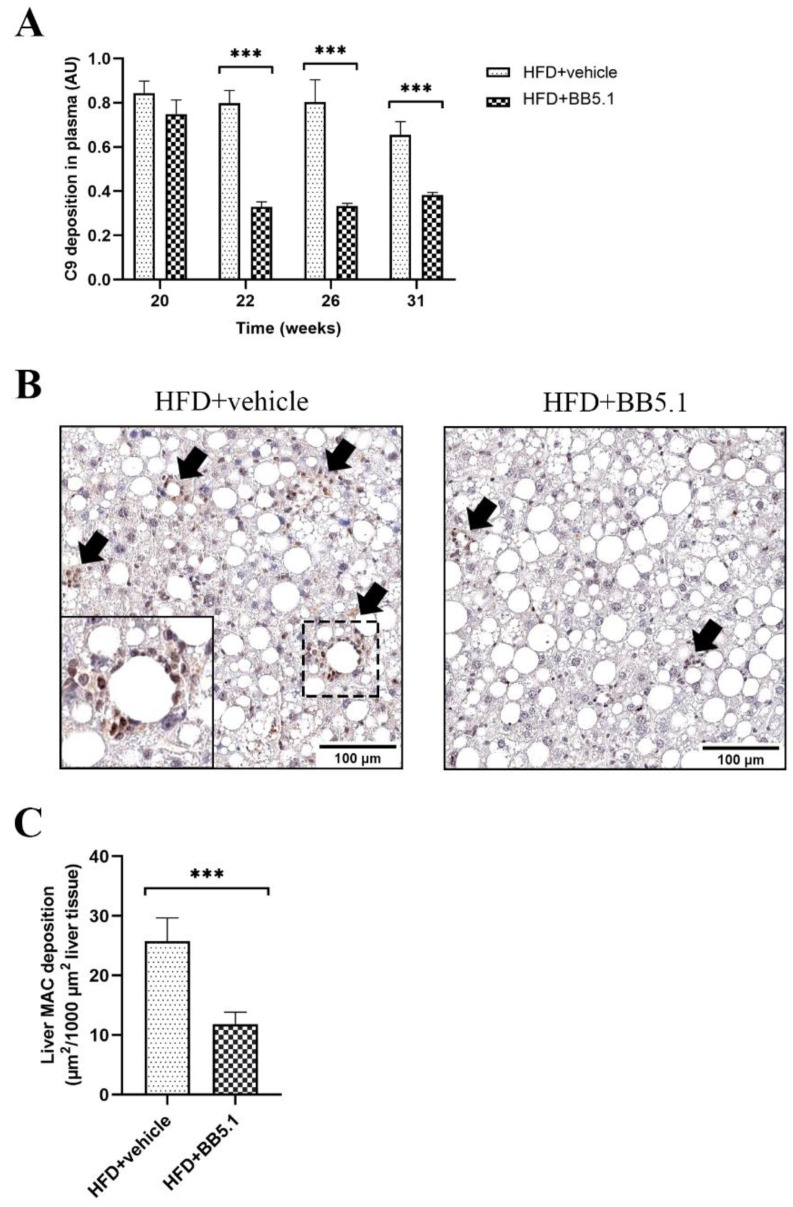
Anti-C5 treatment reduces plasma complement activity and in situ complement activation in liver. (**A**) Anti-C5 treatment reduced Ig-M stimulated C9 deposition in plasma. (**B**) Representative pictures of MAC immunostaining from HFD+vehicle and HFD+BB5.1 livers, respectively. Positive staining is indicated by arrows. (**C**) Anti-C5 treatment reduced liver MAC deposition. Data are mean ± SEM. *** *p* ≤ 0.001 HFD+vehicle vs. HFD+BB5.1.

**Figure 3 ijms-23-10736-f003:**
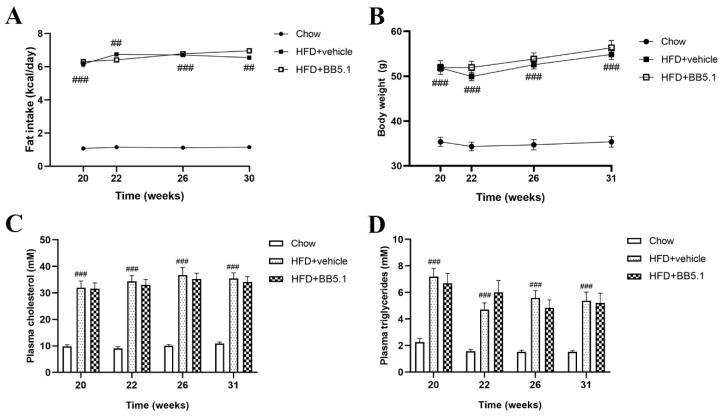
Anti-C5 treatment does not affect HFD-induced obesity and its metabolic risk factors. (**A**) Fat intake, (**B**) body weight, (**C**) plasma total cholesterol and (**D**) triglyceride concentrations are increased by HFD feeding and not altered by the anti-C5 treatment. Data are mean ± SEM. ## *p* ≤ 0.01, ### *p* ≤ 0.001 Chow vs. HFD+vehicle.

**Figure 4 ijms-23-10736-f004:**
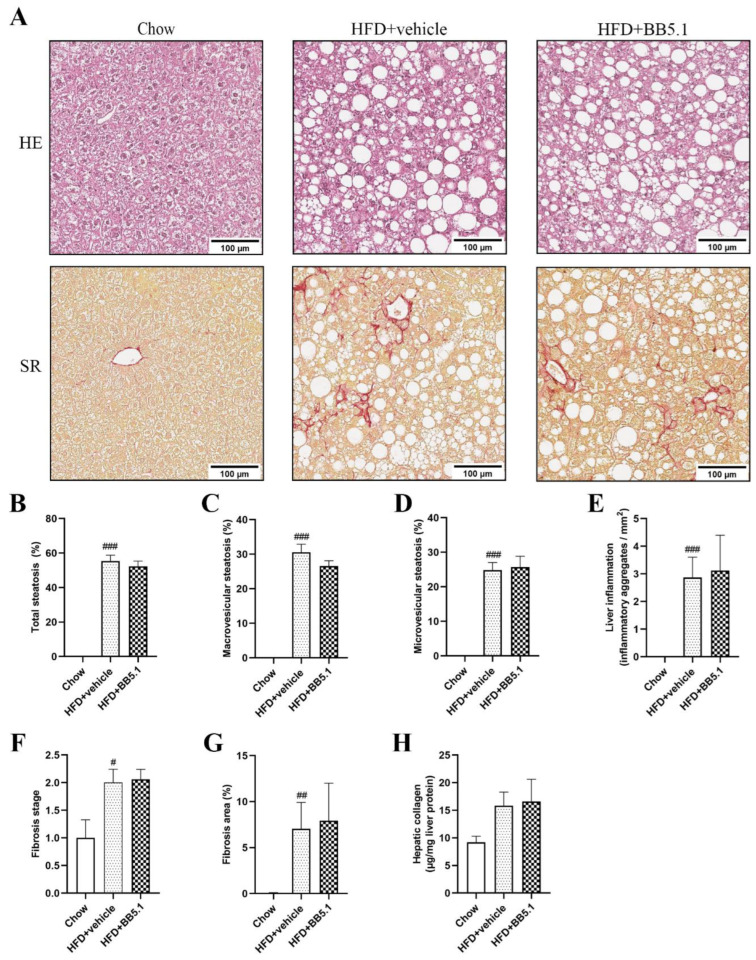
Anti-C5 treatment does not affect NAFLD/NASH and liver fibrosis. (**A**) Representative pictures of livers stained with Hematoxylin-Eosin (HE) and Sirius Red (SR) from Chow, HFD+vehicle and HFD+BB5.1 animals, respectively. (**B**) Liver steatosis, including (**C**) macrovesicular and (**D**) microvesicular steatosis, and (**E**) inflammation are induced by HFD feeding and not affected by the anti-C5 treatment. Fibrosis (**F**) stage and (**G**) area are also increased by HFD feeding but not altered by the anti-C5 treatment. (**H**) Hepatic collagen was not affected by anti-C5 treatment. Data are mean ± SEM. # *p* ≤ 0.05, ## *p* ≤ 0.01, ### *p* ≤ 0.001 Chow vs. HFD+vehicle.

**Figure 5 ijms-23-10736-f005:**
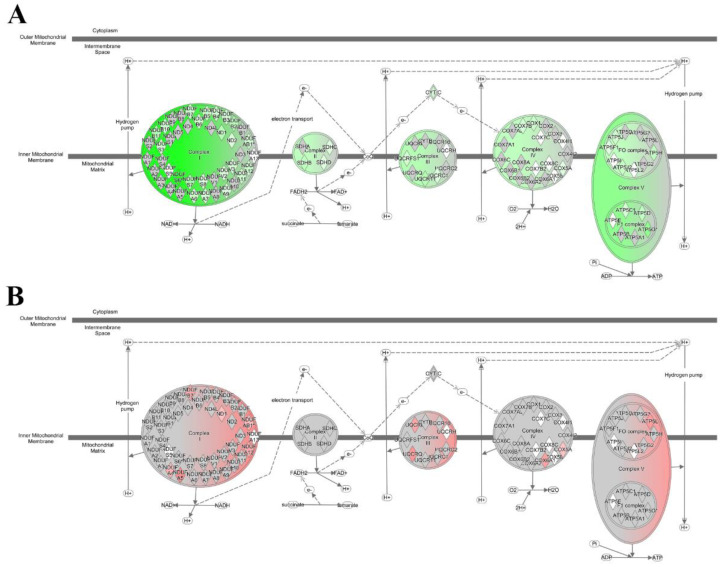
Anti-C5 treatment improves oxidative phosphorylation canonical pathway. (**A**) HFD feeding downregulates many genes involved in oxidative phosphorylation pathway, comparison: HFD+vehicle vs. Chow while (**B**) anti-C5 treatment upregulates a subset of oxidative phosphorylation genes, comparison: HFD+vehicle vs. HFD+BB5.1. Green: gene downregulation, red: gene upregulation.

**Figure 6 ijms-23-10736-f006:**
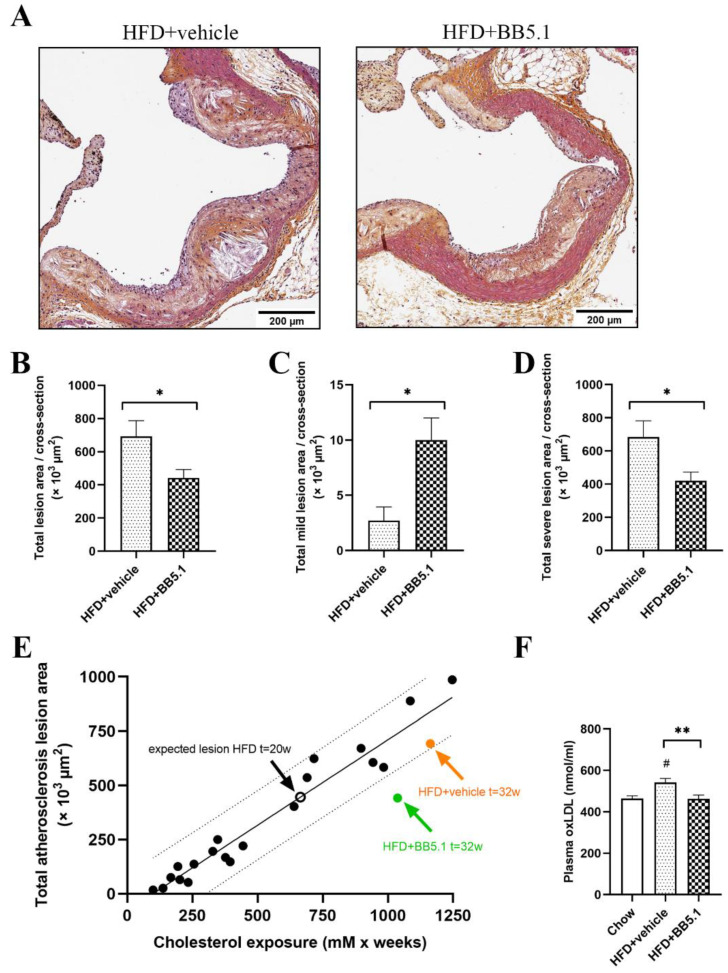
Anti-C5 treatment reduces atherosclerotic lesion size and severity. (**A**) Representative pictures of atherosclerotic plaque from HFD+vehicle and HFD+BB5.1 aortas, respectively. (**B**) Anti-C5 treatment reduces the total lesion area, (**C**) increased the area containing mild lesions and (**D**) decreased the lesion area containing severe lesions. (**E**) Linear relationship between cholesterol exposure and total lesion area in Ldlr-/-.Leiden mice based on control group data from historical studies, dashed lines indicate the 95% prediction band. (**F**) Plasma oxLDL is increased by HFD feeding and was reduced by anti-C5 treatment back to Chow level. Data are mean ± SEM. # *p* ≤ 0.05 Chow vs. HFD+vehicle. * *p* ≤ 0.05, ** *p* ≤ 0.01 HFD+vehicle vs. HFD+BB5.1.

**Figure 7 ijms-23-10736-f007:**
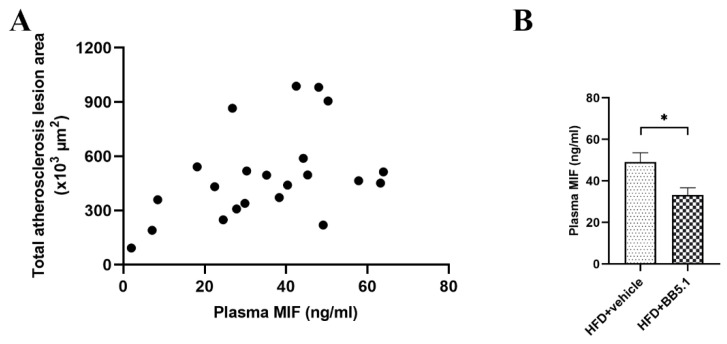
The levels of circulating MIF correlate with the size of atherosclerosis lesion and are reduced with anti-C5 treatment. (**A**) Spearman correlation between MIF concentration in plasma and the size of atherosclerosis plaque (*p* = 0.033, ρ = 0.47). (**B**) Anti-C5 treatment reduced plasma MIF. Data are mean ± SEM. * *p* ≤ 0.05 HFD+vehicle vs. HFD+BB5.1.

## Data Availability

The transcriptomics dataset generated for this study can be found in the Gene Expression Omnibus (GEO) repository (https://www.ncbi.nlm.nih.gov/gds), under accession number: GSE195614.

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
