# Peer review of "Therapeutic Intervention with Anti-Complement Component 5 Antibody Does Not Reduce NASH but Does Attenuate Atherosclerosis and MIF Concentrations in Ldlr-/-.Leiden Mice"

_ijms, 2022, doi:10.3390/ijms231810736_

Round 1
Reviewer 1 Report
In this paper, Seidel. et al report the therapeutic effect of anti-C5 ab for treatment of atherosclerosis. The study is very important. The finding is interesting. My comments are the following:
1) Previously, Wu et al (circ, res. 2009, 104, 550) used anti-C5 ab together with use of mCD59ab and Apoe-/- mice showing the beneficial effect on preventing atherosclerosis. The findings reported in Wu et al's paper rationalize well for current study. Please introduce this work in the introduction to justify for the current study.
2) Please perform the complement and MAC staining to show if there are any differences in complement deposition of the plaque in aorta. This additional result would highlight the importance role of the inhibition of MAC and C in preventing atherosclerosis progression.
3) They also show that antii-C5 antibody treatment inhibits the serum level of ox-LDL. This finding is comparable the observation published by Dai, et al (Frontier, Cardiovscular. Med. 2021, 8). Please discuss these finding.
Reviewer 2 Report
Title: Therapeutic intervention with anti-complement component 5 antibody does not reduce NASH progression but does attenuate atherosclerosis development in Ldlr-/-.Leiden mice.
The authors investigated the role of component 5 of innate immunity in non-alcoholic steatohepatitis (NASH) and atherosclerosis induced by HFD in Ldlr-/-.Leiden mice. They found that anti-C5 treatment reduced complement system activity in plasma and hepatic membrane attack complex deposition but failed to protect from NASH; however, treatment was effective against established atherosclerosis. This is a novel and interesting study; however, I have some concerns.
1. Explain the advantages and disadvantages of the Ldlr-/-.Leiden mice model of NASH in detail. Is it similar to human NASH? There are several experimental models of NASH, why this one and no other?
2. What is the role that other important proinflammatory signaling pathways play in NASH progression, including NF-kappaB and NLRP3 inflammasome signaling pathways? Are they more important than component 5 of innate immunity?
3. If chronic inflammation is an important driver in the progression of NASH and atherosclerosis, then proinflammatory mediators (cytokines) must be measured in experimental groups.
Round 2
Reviewer 2 Report
Thank you for improving your manuscript according to the comments of the reviewers.